# Sequential buckling in fluid-filled cylindrical shells
Shresht Jain [1,2], Finn Box [1,2], Martin Quinn[1,2], Chris Johnson [2,3] & Draga Pihler-Puzović [1,2] ✉

From oil drums to flying rockets, cylindrical shells are valued for their load-carrying capacity. When sufficiently compressed, they buckle, with the phenomenon taking many forms, from periodic diamond-shaped buckles to localized elephant footing. The precise physical mechanisms of buckling are different, for example, in empty shells and shells with a solid core. However, despite the abundance of liquid-filled shells in industry and everyday life, their buckling is largely overlooked. Here, we compress beverage cans and identify a sequential buckling instability that localizes circumferential rings above a critical level of compression. Combining measurements of the anisotropic material properties of the shell with modelling based on the nonlinear Swift-Hohenberg equations, we demonstrate that fluid-filled shells can support a multiplicity of localized solutions, which are induced by the nonlinear hoop stress of the shell and sequentialize through homoclinic snaking. This establishes a rare link between idealized mathematical studies of pattern formation and physical realizations of spatially-localized buckling phenomena. These findings serve as a blueprint for exploring localized patterns induced by material nonlinearities, near-incompressibility and pressurization in other physical systems.

The buckling of cylindrical shells underpins a wide variety of phenomena, from the structural failure of rocket fuselages[1] and grain silos[2,3] to the bending of rod-shaped bacteria[4]. Under compressive loads, buckling can force a cylindrical shell to adopt different shapes depending on imperfections[5], boundary conditions, wall thickness and plasticity[6], internal pressurization[7] and the presence of a soft core[8–10], rigid core[11] or mandrel[12]. Theoretically predicted linear thresholds for the onset of instability in cylindrical shells are typically larger than those found in experiments[13]. Hence, the historical focus has centred on determining the 'knockdown factor', by which these thresholds must be empirically reduced to ensure the design of robust structures[14]. Recently, by applying a lateral point indentation to empty, axially-loaded cylindrical shells, Virot et al.[15] demonstrated that a finite-amplitude perturbation was required for catastrophic failure and, in doing so, mapped the border of the basin of attraction for stable, unbuckled fixed points. Building on these findings, numerical investigations of empty and pressurised cylindrical shells under axial compression have shown that diamond-shaped buckles grow from a localized dimple, which can be induced by point indention[16].

The linear stability of pressurized cylindrical shells and cylindrical shells filled with incompressible fluid under axial loading is well-established[17–19], but fails to explain localized buckling. Indeed, under static loading, axisymmetric buckles typically appear instantaneously upon reaching a critical strain[11], and pattern the entire domain with a well-defined wavelength that is selected by the material and geometrical properties of the shell[12,20,21]. (Unless the rim of the can has been altered to have some periodicity, in which case the circumferential symmetry can be broken in a programmable manner[22]). Cylindrical shells undergoing axial crushing have been observed to buckle progressively via the formation of a fold, near the moving boundary, which self-contacts upon further deformation, leading to another buckle forming adjacent to the first buckle[23,24]. As such, this type of sequential buckling is strongly influenced by boundary conditions and inherently requires large strains.

Pertinently, theoretical studies of long beams and inflated plates[25,26], and diamond-shaped (Yoshimura) buckles in cylindrical shells[27,28], have invoked the nonlinear Swift-Hohenberg (SH) equations to describe the appearance of (steady) localized patterns, which propagate outwards with changing control parameter until they fill the domain space with a periodic pattern. Originally proposed as a model for thermal convection[29], the SH equations have been studied extensively in the context of pattern formation[30–32] and are known to exhibit dynamically complex solutions like isolas, solitons and sequential localized solutions that arise through homoclinic snaking[33]. However, aside from theoretical studies that relate the

[1]Physics of Fluids & Soft Matter, Department of Physics & Astronomy, University of Manchester, Manchester, UK. [2]Manchester Centre for Nonlinear Dynamics, University of Manchester, Manchester, UK. [3]Department of Mathematics, University of Manchester, Manchester, UK.
✉e-mail: draga.pihler-puzovic@manchester.ac.uk

progressive pattern formation predicted by the SH equations to specific physical systems, experimental realizations of the mechanism are limited[34–36]. It is also worth noting that commercial beverage cans have nonlinear[37] and anisotropic material properties, which arise as a consequence of the die-forming process that shapes them, and material non-linearities are known to cause localisation of buckling phenomena in, e.g., curved films on compliant substrates[38], beams on an elasto-plastic foundations[39] and long, thick cylindrical shells under various loading conditions[40–43].

Here we demonstrate that under axial loading, liquid-filled cylindrical shells, including those that are pressurized, exhibit a sequence of spatially-localized ring buckling events that pattern the shell sequentially, as evidenced by the compression of a can in Fig. 1a. We develop a theoretical framework that reveals the mechanism that underpins this localized, axisymmetric buckling by integrating nonlinear constitutive behaviour into a physical model, based on a SH equation for the normal displacement of the shell. Our model predicts spatially-localized surface patterns and a proliferation of buckles upon increasing compression until the entire shell domain is covered. Through direct comparison with experiments, we thus demonstrate that the sequential emergence of ring buckles is a consequence of nonlinear, and spatially anisotropic, hoop stresses that soften and re-stiffen in response to axial compression and internal pressurization. In doing so, we present a means of die-less hydro-forming that can pattern pre-filled beverage cans with axial corrugations.

## Methods
### Experiments
Compression tests were performed on a variety of commonly available beverage cans with radii $26.6 \leq R \leq 43$ mm, thicknesses $0.09 \leq t \leq 0.20$ mm, and length $115 \leq L \leq 190$ mm. The samples were compressed axially in a Universal Testing System (Instron 3345) at a fixed speed $\delta \dot{x}$ (which resulted in an end shortening of $\delta x = T \delta \dot{x}$, where $T$ is the time since the beginning of the test) and measuring the corresponding force $F$. Tests were performed at different compression speeds ranging from $0.1 \leq \delta \dot{x} \leq 8.5$ mm s$^{-1}$, and showed no dependence on compression speed. During compression, can profiles were imaged with Nikon D90 and D7100 cameras, recording at a frame-rate of 25 fps. The images were processed using MATLAB, to obtain the profile outlines as shown in Fig. 1a. We also extracted the peak-to-peak

separation distance between the buckles for each sample before catastrophic failure, and computed the pattern wavelength for each test. For each geometry, these wavelengths were further averaged to yield the average wavelength $\langle \lambda \rangle$, with the error on $\langle \lambda \rangle$ corresponding to the standard deviation of data from up to five geometrically identical samples. All tests were carried out with unopened soda cans with an initial internal pressure $Q_i$ in the range of $2 - 4$ atm[44], or cans that were drained, refilled with water and resealed at 1 atm. These were classified as pressurised and unpressurised tests, respectively. Despite the significant difference in the initial values of the pressurization, the same phenomena were observed in all tests, indicating that the near-incompressibility of the shell content was more important for the appearance of ring buckles. The sequential buckling also occurred at much smaller strains and without self-contact, implying that a mechanism different to axial crushing underpins the post-buckling behaviour of fluid-filled shells, such as in Fig. 1a.

### Modelling
To understand the essential mechanism for sequential buckling, we formulate a pseudo-plastic model for the deformation of the fluid-filled cans, modelling them as shallow cylindrical shells of radius $R$, length $L$ and thickness $t$. We assume that shell deformations $w$ are small relative to its radius and remain axisymmetric. Following[45], we obtain the force and moment balance equations

$$\frac{dN_x}{dx} = 0, \tag{1}$$

$$\frac{d^2 M_x}{dx^2} + \frac{N_\theta}{R} - N_x \frac{d^2 w}{dx^2} = Q_L, \tag{2}$$

where $N_x$ and $N_\theta$ are force intensities (i.e., forces per unit perimeter) in axial and circumferential directions, $M_x$ is the axial bending moment intensity, and $Q_L$ is the internal pressure, see Supplementary Note 3 for more details.

Assuming that compressive loading supplies all of the axial force intensity, we obtain

$$N_x = -\frac{F}{2\pi R} \tag{3}$$

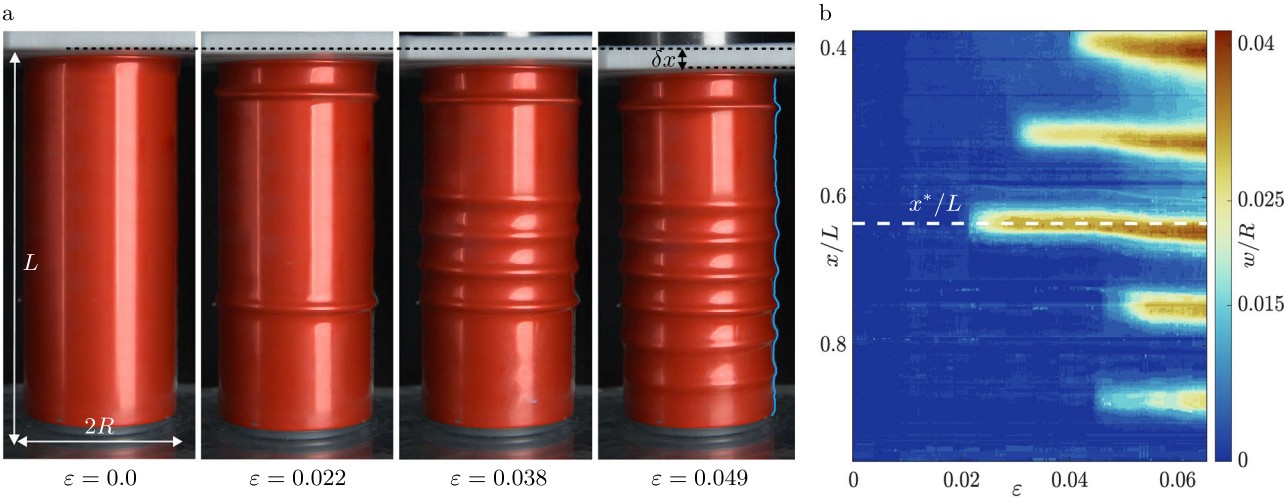

**Fig. 1 | Sequential buckling. a** Deformation of a fluid-filled beverage can under increasing applied engineering strain $\varepsilon = \delta x / L$, with the first buckle appearing at $\varepsilon = 0.02$ and the can losing integrity at $\varepsilon = 0.064$. The undeformed can is of radius $R = 26.5$ mm, axial length $L = 137$ mm and wall thickness $t = 0.1$ mm. Image processing techniques, described in Supplementary Note 1, are used to find the surface profile of the can (highlighted in cyan). The images have been edited in Adobe Photoshop to remove any trademarks without altering the phenomenon.

**b** Evolution of the can profile highlighted in (**a**): scaled displacements from initial radius $w/R$ are represented using the color scheme. The first buckle appears near the middle of the can at $x^\star/L \approx 0.6$, and grows to amplitude $w_{max}/R \approx 0.04$, with subsequent buckles radiating axially. The saturation amplitude is consistent to within 8% between samples of the same geometry but varies for different geometries, see Supplementary Note 2.

from the force balance equation (Eq. (1)). The azimuthal and axial force intensities are linked via

$$N_\theta = \nu N_x + (1-\nu^2)C_\theta(\varepsilon_\theta)\varepsilon_\theta, \quad (4)$$

where $\nu$ is the Poisson's ratio and the second term on the right-hand side corresponds to the force intensity due to hoop stresses. To model the role of pseudo-plasticity, we follow the approach of[46] and capture the nonlinear constitutive response of the material by assuming that the coefficient $C_\theta = C_\theta(\varepsilon_\theta)$ is a nonlinear function of the circumferential strain $\varepsilon_\theta = w/R$. To determine $C_\theta$, we subjected rectangular cut-outs of the shell wall in the circumferential direction to extensional tests (for details, see Supplementary Note 4), and fitted

$$C_\theta = C(1 + \gamma_1\varepsilon_\theta + \gamma_2\varepsilon_\theta^2), \quad (5)$$

to the acquired experimental data to obtain the extensional stiffness $C$, and coefficients $\gamma_1$ and $\gamma_2$.

Similarly, we assume that the axial bending moment in Eq. (2) is a nonlinear function of the axial curvature $\kappa_x = \frac{d^2w}{dx^2}$. To determine $M_x$, we performed three-point-bend tests on rectangular cut-outs of the shell wall in the axial direction (for details, see Supplementary Note 4), and fitted

$$M_x = D(\kappa_x + \alpha\kappa_x^2), \quad (6)$$

to the acquired experimental data to determine the bending stiffness $D$ and coefficient $\alpha$. For the cans used in measurements in Fig. 2a, $C = 860 \pm 32\,\text{N mm}^{-1}$, $\gamma_1 = -2.72 \pm 0.17$, $\gamma_2 = 2.96 \pm 0.43$, $D = 102.15 \pm 0.59$ N mm and $\alpha = -12.84 \pm 0.19$ mm, assuming $\nu = 0.35$ as appropriate for aluminium. We stress that the polynomial relationships in Eqs. (5) and (6), and hence the values of these coefficients, are empirical and have been chosen to obtain a tractable reduced model.

Combining Eqs. (1)–(6) results in a 1D beam equation that resembles the form of the SH equation:

$$D\left[\frac{d^4w}{dx^4}\left(1 + 2\alpha\frac{d^2w}{dx^2}\right) + 2\alpha\left(\frac{d^3w}{dx^3}\right)^2\right]$$
$$+ \frac{F}{2\pi R}\frac{d^2w}{dx^2} - \nu\frac{F}{2\pi R^2} + \frac{(1-\nu^2)C}{R}\left(\frac{w}{R} + \gamma_1\left(\frac{w}{R}\right)^2 + \gamma_2\left(\frac{w}{R}\right)^3\right) = Q_L. \quad (7)$$

The choice of nonlinear functions in (5) and (6) can be thought of as an expansion of the hoop-stress stiffness and the axial bending moment intensity to higher orders in the circumferential strain. However, because of this choice, the quadratic term in the hoop stresses (with the multiplier $\gamma_1$) is negative while the cubic term in the hoop stresses (with the multiplier $\gamma_2$) is positive in Eq. (7). In the SH equations, the competition between such softening and re-stiffening terms leads to localization and homoclinic snaking, and we will demonstrate that the same nonlinear dynamics is responsible for the sequentialization of buckling here.

The model (Eq. (7)) is closed by enforcing Boyle's law, which determines the pressure $Q_L$ (see Supplementary Note 5). Assuming $\mathcal{I}$ is the fraction of the initial can volume filled with an incompressible fluid, while the rest is filled with an ideal gas, allows us to write an integral constraint that expresses the change of volume during compression as

$$Q_L\left[\int_0^{L-\delta x}\pi(w+R)^2\,dx - \mathcal{I}\pi R^2 L\right] = Q_i[1-\mathcal{I}]\pi R^2 L. \quad (8)$$

Thus, by changing the parameter $\mathcal{I}$, we can vary the compressibility of the can: a value of 1 corresponds to a fully-filled i.e., incompressible can, while for $\mathcal{I} < 1$, any change in volume is accompanied with the corresponding change in the internal pressure $Q_L$. In experiments, this corresponds to the compressibility of the can due to a small volume of air trapped inside, which we measured by opening and emptying beverage cans and comparing the contents with the volume of liquid required to fully fill the emptied cans, finding $\mathcal{I} \approx 0.9$.

We relate the force $F$ to the level of compression $\delta x$ by assuming that the can does not stretch in the axial direction, which leads to another integral constraint:

$$\int_0^{L-\delta x}\sqrt{1 + \left(\frac{dw}{dx}\right)^2}\,dx = L. \quad (9)$$

Equations (7)–(9) were solved using parameter continuation in AUTO-07p[47] via a continuation performed in the applied force $F$. Our implementation included mapping the integration domain to a fixed interval and rewriting the fourth-order differential equation into four first-

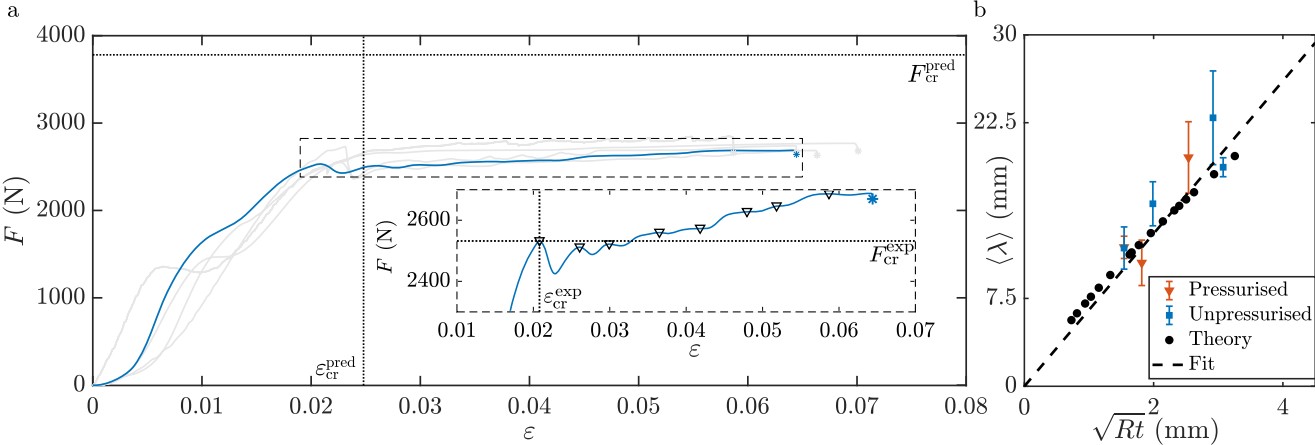

**Fig. 2 | Measuring buckling. a** Measured force $F$ as a function of imposed engineering strain $\varepsilon$ in compression experiments using samples such as in Fig. 1a ($R = 26.5$ mm, $L = 137$ mm and $t = 0.1$ mm). Tests, each corresponding to a different $F - \varepsilon$ line, were performed at strain rates ranging from $\sim 10^{-4}\text{s}^{-1}$ to $\sim 0.06\,\text{s}^{-1}$; results are independent of strain rate. Dotted lines indicate predictions from our model, Eqs. (7)–(9), for the force ($F_{cr}^{pred} = 3840N$) and strain ($\varepsilon_{cr}^{pred} = 0.0283$) at which the first buckle forms. Inset: a detailed view of the stress-strain curve for the highlighted test (in blue) chosen randomly from the $F - \varepsilon$ data; the appearances of sequential buckles and catastrophic failure are indicated using triangles and the (blue) star,

respectively. The critical values of strain ($\varepsilon_{cr}^{exp} = 0.021$) and force ($F_{cr}^{exp} = 2536.8$ N) measured at the onset of buckles for this experiment are also shown with dotted lines. These are measured for all tests in (**a**) and averaged to obtain $\langle\varepsilon_{cr}^{exp}\rangle$ and $\langle F_{cr}^{exp}\rangle$ shown in Fig. 3f. **b** The measured peak separation $\langle\lambda\rangle$ as a function of the geometric mean of the can radius and thickness. Markers represent initially pressurised (squares) and unpressurised (triangles) cans and predictions of our model, Eqs. (7)–(9), where $5 \leq R \leq 106$ mm (dots). The dashed line indicates a linear fit on the experimental data $\langle\lambda\rangle = \mathcal{A}\sqrt{Rt}$, with fitted $\mathcal{A} = 6.47 \pm 1.78$ and a coefficient of determination equal to 0.803.

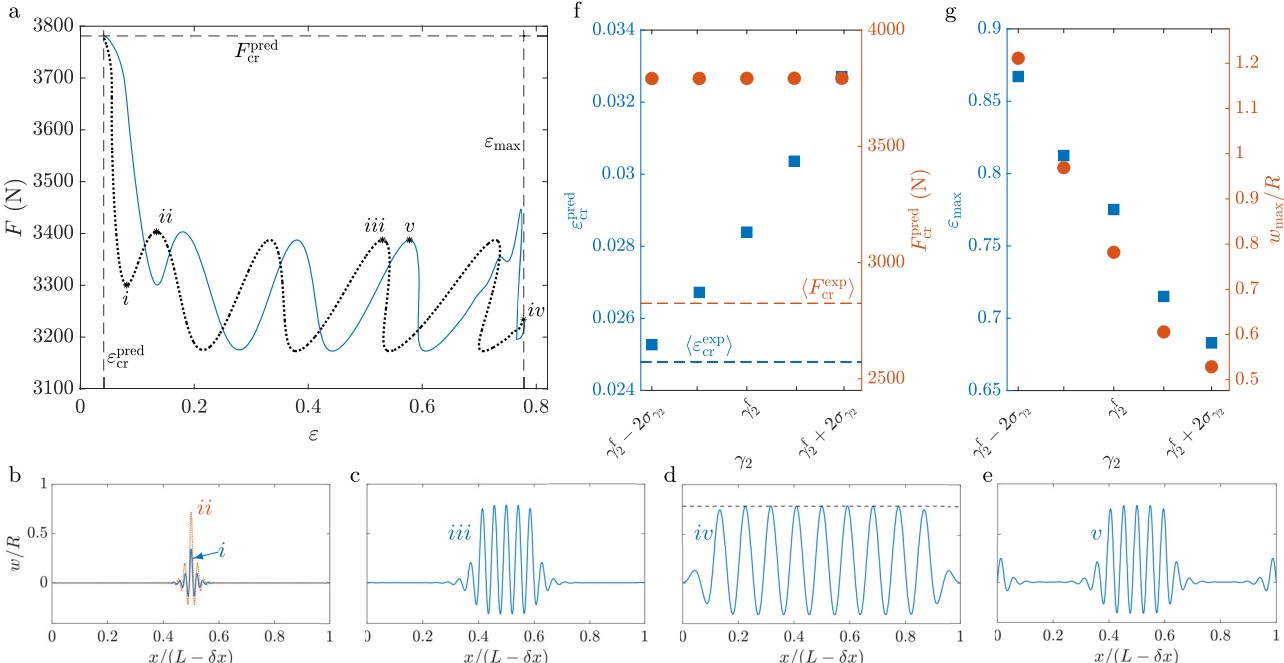

**Fig. 3 | Modelling buckling. a** Force $F$ as a function of the engineering strain $\varepsilon$ predicted numerically from Eqs. (7) to (9); (blue) solid and (black) dotted lines correspond to the buckled solution branches with and without undulations near the can edges, respectively. Can deformations $w/R$ that correspond to points labelled with *i-v* are shown in (**b-e**), respectively, with $w_{max}/R$ indicated in **d** with the dashed line: **b** $\varepsilon = 0.0815$ and $F = 3300.5$ N for (blue) solid profile and $\varepsilon = 0.134$ and $F = 3403.1$ N for (red) dotted profile, **c** $\varepsilon = 0.531$ and $F = 3387.4$ N, **d** $\varepsilon = 0.776$ and $F = 3234.6$ N, **e** $\varepsilon = 0.578$ and $F = 3387.4$ N. The dashed lines indicate significant quantities, which include the critical strain $\varepsilon_{cr}^{pred}$ and force $F_{cr}^{pred}$, respectively, which are measured at the point when the localised solutions appear and compared to the experimental onset of buckling in (**f**), and the maximum strain $\varepsilon_{max}$ for which solutions have been found. (**f**) Variation in the predicted critical values of the engineering strain $\varepsilon_{cr}^{pred}$ and force $F_{cr}^{pred}$ with the multiplier $\gamma_2$ from Eq. (5), where $\gamma_2^f = 2.96$ is the fitted value of $\gamma_2$ and $\sigma_{\gamma_2} = 0.43$ is the corresponding uncertainty in this fitted parameter, see Supplementary Note 4. The corresponding experimental values from Fig. 2a are indicated using dashed lines. **g** Variation of the maximal computed strain $\varepsilon_{max}$, as defined in (**a**), and the maximal amplitude of undulations $w_{max}$ with $\gamma_2$.

order equations, which were then solved by the method of orthogonal collocation as is the default in AUTO-07p[47]. For more numerical details, see Supplementary Note 6.

## Results

Compressed beverage cans visibly deform in the radial direction relative to the start of the experiment, but the extent of this deformation varies along the length of the cans, see the profile outline highlighted in Fig. 1a. To showcase the evolution of the profiles in Fig. 1a with increasing strain, the measured radial deformation scaled by the can radius, $w/R$, along the cylindrical axis (where $x$ corresponds to the distance from the top of the undeformed can), is shown in Fig. 1b as a function of engineering strain defined as $\varepsilon = \delta x/L$, where $\delta x$ is the end shortening and $L$ is the length of the undeformed can.

From the evolution of the deformation profiles, we see that a series of axi-symmetric buckles occurs sequentially. The location at which the first buckle appears ($x^\star$ in Fig. 1b) is typically close to the middle of the can, but the exact location changes from test to test, and is believed to be sensitive to small imperfections in the can geometry and stiffness. The amplitude of an individual buckle grows to a maximum $w_{max}$, whereafter a new buckle starts to appear. Subsequent buckles grow adjacent to pre-existing ones, following the same cycle of growth, amplitude saturation and buckle nucleation until the entire length of the can is filled with buckles. The formation of buckles plastically deforms the can, and the buckles remain when the loading is removed. Hence, the original length of the can, and its undeformed profile cannot be recovered post-loading, so a new can was used for each experimental test.

Measured values of force as a function of engineering strain for cans of the same geometry are shown in Fig. 2a. Data from five different samples are presented, with the highlighted measurement also shown in a detailed view as the inset. Comparing these measurements to the can profiles at the

corresponding engineering strains, we observe that the appearance of the first buckle corresponds to a sharp decrease in the recorded force. As the buckle subsequently grows in amplitude, the recorded force also increases, until the next buckle appears with another decrease in the recorded force and so on. The experiment ends with the whole surface of the can being covered by buckles, which is then followed by catastrophic failure (the can bursting) as the level of compression increases further, indicated by the star markers in Fig. 2a.

In Fig. 2b, we plot the average wavelength for each geometry $\langle \lambda \rangle$ against the characteristic lengthscale given by the geometric mean of the radius and the thickness of a cylindrical shell, $\sqrt{Rt}$[13,17]. We find that the average wavelength scales linearly with $\sqrt{Rt}$, as shown by the fitted line in Fig. 2b given by $\langle \lambda \rangle = \mathcal{A}\sqrt{Rt}$, where the pre-factor $\mathcal{A} = 6.47 \pm 1.78$. This scaling resembles the scaling observed in classical literature on the wrinkling of linear elastic shells with internal pressure[17].

In Fig. 3a, we show the predicted $F$ as a function of engineering strain $\varepsilon$ by analogy with Fig. 2a (and for the same parameters). We find multiple solution branches, each associated with monotonic variation of $F$ with $\varepsilon$, which are attained via homoclinic snaking. These solutions are disconnected from the trivial (pre-buckling) branch, which is omitted from Fig. 3a for clarity. We also plot a few examples of radial deformation $w$, which belong to different branches of solution at similar values of $F$, see Fig 3b–e. Homoclinic snaking is hard to trace numerically (see Supplementary Note 6), so we constrain our solutions in Fig. 3 to be left-right symmetric, though isolated solutions that break this symmetry have been computed as well. All solutions look like periodic waves modulated by an envelope that localises them.

Moving from left to right in Fig. 3a, solutions on the first branch, for which $F$ decreases with increasing $\varepsilon$, have a single prominent undulation, and resemble the first buckle seen experimentally (see Fig. 3b). The critical force and strain where the branch originates are marked with $F_{cr}^{pred}$ and $\varepsilon_{cr}^{pred}$, respectively, and compared to experimental measurements in Fig. 2a. Our

model, which relates the applied force to the end-shortening of the can through the assumption of inextensibility, is limited in capturing the (nonlinear) deformation, which occurs before buckling, shown in Fig. 2a, see the curves for $\varepsilon$ between 0 and $\sim 0.2$. Nevertheless, the agreement between the experiments and predictions from the theory is within 33% for both values of the critical force and the strain at the onset of buckling (see also the discussion of Fig. 3f, g below). Continuation in the force towards larger values of $\varepsilon$ along this branch increases the amplitude of the first buckle, see Fig. 3b. This continues until its amplitude saturates at $w_{max}$ at the point marked *ii* in Fig. 3a, where another solution branch originates.

Beyond this point in Fig. 3a, with each new solution branch, the width of the envelope increases, so more undulations fit inside the envelope, see, e.g., Fig. 3c. The amplitude of these localized waves increases with increasing $\varepsilon$, until each newly growing undulation saturates at the maximum amplitude given by $w_{max}$. Once all undulations have reached the maximum, a new solution branch is created. These solutions, therefore, resemble can profiles with ever-increasing number of buckles. Hence, solutions with three, five, seven etc undulations are seen, until waves cover the entire surface of the can (Fig. 3d). The predicted peak-to-peak separations from our theoretical model are in agreement with the values of $\langle \lambda \rangle$ obtained in experiments, see Fig. 2b. Note that the peak-to-peak separation in the predicted patterns remains constant to within 1% as buckles appear; apparent variations in Fig 3b–e are a consequence of presenting profiles in the coordinate system of the can, given by $x/(L - \delta x)$. Hence, the wavelength appears to change as $\delta x$ changes.

Once the entire domain is filled with buckles, continuation in the loading force leads to solutions in which the strain decreases again. The largest strain, for which solutions are computed in Fig. 3, is denoted as $\varepsilon_{max}$. The new solutions, which appear as parallel lines to the previously discussed solutions in the force-strain bifurcation plot in Fig. 3a, are also localized. A typical radial displacement $w$ associated with these solution branches is shown in Fig. 3e. As before, it resembles a localized wave in the middle of the domain, but with additional undulations at its edges. Given similarities between Eq. (7) and the SH equations, there are likely many more solutions that are variants of the localized states shown in Fig. 3, and are attained via homoclinic snaking[27]. In experiments, perturbations to the can stiffness and geometry can initialise any of these states at a given level of compression and are presumably responsible for e.g., some randomness in the position of the first buckle within the domain and occasional simultaneous appearance of buckles near the can boundary and its middle; see also the Supplementary Video of cans deformed at different compression rates.

The results presented in Fig. 3a are comparably sensitive to the values of fitted coefficients, $\gamma_1$ and $\gamma_2$, used in the nonlinear relationship Eq. (5). By comparison, the nonlinearities due to terms containing $\alpha$ in Eq. (7) are less significant quantitatively as well as qualitatively. We also investigated the influence of initial can pressure $Q_i$ on the phenomena, noting that while it had a non-negligible quantitative effect, the qualitative phenomena remains unchanged despite increasing $Q_i$ from 0 atm (corresponding to unpressurised cans) to 2 atm (corresponding to pressurised cans). Here we show how the results vary with $\gamma_2$, while the other variations are presented in more detail in Supplementary Note 7.

From analogy with the SH equations, we expect that the softening response due to the quadratic nonlinearities (i.e., with the multiplier $\gamma_1$) makes the shell easier to deform and buckle once the initial undulation occurs, while the re-stiffening induced by the cubic nonlinearities (i.e., with the multiplier $\gamma_2$) makes it energetically favourable to create a subsequent buckle once the amplitude of an undulation is close to $w_{max}$. To demonstrate these effects, we repeated the analysis from Fig. 3a for $\gamma_2^f - 2\sigma_{\gamma_2} \leq \gamma_2 \leq \gamma_2^f + 2\sigma_{\gamma_2}$, where $\sigma_{\gamma_2}$ is the uncertainty in the fitted parameter $\gamma_2$ and $\gamma_2^f$ is its fitted value. The resulting critical force and strains are shown as a function of $\gamma_2$ in Fig. 3f, where we also indicate the values found experimentally.

Varying $\gamma_2$ also affects the predictions for $w_{max}$ and $\varepsilon_{max}$; both decrease with increasing $\gamma_2$, as shown in Fig. 3g. Hence, decreasing the value of $\gamma_2$ predicts buckles of larger amplitude and requires larger levels of compression

before buckles populate the entire surface of the can. The relatively large predicted compression is a direct consequence of assuming inextensibility in Eq. (9). Indeed, theoretical predictions of $w_{max}$ and, consequently, $\varepsilon_{max}$ tend to be larger than that observed in experiments (see e.g., Fig. 1b).

## Discussion

We have demonstrated that uni-axial compression of fluid-filled cylindrical shells can result in the sequential formation of corrugations in the shell wall. This sequential buckling is markedly different from the catastrophic collapse of an empty can subject to compression and point indentation[15]. Distinct nonlinearities in the hoop stress, which act to soften and then re-stiffen the shell, also render this sequential buckling phenomenologically different from the periodic ring buckles that form simultaneously in a soft shell under compression[11]. Many materials, for example, steel, have a similar constitutive response, where the softening that occurs post-yield is followed by a re-stiffening under increased strain[48]. Hence, we anticipate that the phenomena reported here are not unique to the aluminium shells used in our experiments.

The nonlinear framework of the SH equation applied here for fluidfilled cans is distinct from previous works that have used a similar framework for examining the buckling of cylindrical shells[15,16,27,28] which find nonaxisymmetric buckles with the form of Yoshimura-like diamonds. In the corresponding models, nonlinearities arise from the response of the system to point indentation, whereas in our model, multiplicity, and hence sequentiality, are primarily underpinned by material nonlinearities. It is likely that the die-forming process results in material properties that vary along the length of the can, which is not accounted for in our model; inhomogeneous material properties are known to influence pattern formation[49]. Instead, our model predicts small amplitude peaks that bound the large amplitude localised buckles; these are not observed in the experiments and warrant further investigation. Although the model qualitatively captures localization of the axisymmetric buckles, the predicted maximum amplitude of the buckles is significantly larger than its experimentally measured value, which inevitably leads to a significant difference in the maximum strains as well, via the assumption of inextensibility. It is possible to tune the strength of nonlinearities in our model, via the values of $\gamma_1$ and $\gamma_2$, to improve quantitative agreement between its predictions and the experimental results. However, the real strength of the model is in revealing the underlying mechanism for sequential buckling in a cylindrical shell.

We note that the surface-patterning of a curved elastic bilayer has also been described using a generalized SH equation[50], which revealed a bistable phase that resulted from the presence of dissipative nonlinearities in the interaction between the membrane and the substrate. Similarly, while previous work[16,28] examined the localisation of diamond-shaped buckles on cylindrical shells, and proposed an underlying SH-type mechanism for the destabilisation of the shells to dimple-like perturbations, the nonlinearities required for their analogy to the SH equations came from the nonlinear force response of the shell to point indentation, which is based on experiments described in ref. 15. In contrast, we demonstrate that multiplicity of buckled states can emerge from a competition between softening and restiffening nonlinearities of the shell material, permitting localized solutions to emerge sequentially via homoclinic snaking, even as rotational symmetry is preserved. Our work is also distinct from ref. 51, who explained the onset of buckling in shells via the inversion of localised dimples, since our study goes beyond the onset of buckling to explore the post-buckling sequentialization of periodic ring buckles.

Sequential buckling has also been observed in the macroscopic delamination of a thin film from an elastic substrate[52], the opening of slits in a metallic Kirigami ribbon[53] and in an elastic fingering instability[54] that occurs when air is injected into an elastomeric solid confined inside a Hele-Shaw cell. However, in each of the aforementioned cases, the significance of the sequentially was overlooked or unexplained. Our work provides a blueprint for revisiting these phenomena as an example of spatially-localized pattern formation and searching for the material nonlinearities responsible for sequentiality.

Finally, we note that corrugations increase the structural rigidity of cylindrical shells relative to shells with flat walls. However, the beaded cans often used to transport and store liquid goods are manufactured from thin sheets that are shaped, rolled and joined pre-filling operations. Our findings demonstrate the feasibility of exploiting the material nonlinearities of the shell to cold-form a cylindrical container into a corrugated shape after the filling operation has taken place and without the need for a mould, although a large-scale production process that reduces manufacturing steps by leveraging buckling requires further investigation—especially given the tendency of buckled cans to fail catastrophically through explosive depressurization.

## Data availability

The data that support the findings of this study are available from the following Github folder: https://github.com/shresht-jain/SequentialBucklingProject/.

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

## Acknowledgements
S.J. would like to acknowledges the support of EPSRC DTP Studentship, UK under Grant No. EP-T517823-1 and FB would like to acknowledge the Royal Society (URF/R1/211730).

## Author contributions
All authors conceived the study, S.J., M.Q., F.B. and D.P.P. designed the experiments, S.J., C.J. and D.P.P. developed the modelling, S.J. and F.B. performed experiments, S.J. performed numerical simulations, S.J., F.B., and D.P.P. interpreted the data and wrote the paper.

## Competing interests
The authors declare no conflict of interest. The funders had no role in the design of the study, in the collection, analyses, or interpretation of data, in the writing of the manuscript, or in the decision to publish the results.
