## [Transparent Peer Review file · Communications Physics]

Sequential buckling in fluid-filled cylindrical shells

Corresponding Author: Dr Draga Pihler-Puzovic

Version 0:

Reviewer comments:

Reviewer #1

(Remarks to the Author)

Jain et al. present experiments and modeling of sequential ring buckling in fluid-filled commercial cylindrical shells. The paper is clear and carefully executed, but in its present form it does not sufficiently emphasize what is the new physics. The progression of sequential buckles in soda cans is well known, and the \sqrt{Rt} scaling of the wavelength is trivial from dimensional arguments. The reduced model reproduces the experimental data, as it is reverse engineered to do so: The authors fit nonlinearities from cut-strip tests, insert them into a Swift–Hohenberg–like balance, impose the correct global constraints (Boyle’s law for the headspace, axial inextensibility), and unsurprisingly obtain the observed localization and serrated force–strain curves. If I understand correctly, the real novelty lies in Eq. (5), with the fitted parameters $\gamma_1 < 0$ and $\gamma_2 > 0$. If this is the case, then the article will need revision to convincingly stress that as the main result. In its current form, I do not recommend publication in CommPhys.

Major Comments

1. The main point of the paper is not made clearly. If the real physics is that the nonlinear hoop law [Eq. (5), with $\gamma_1 < 0$ and $\gamma_2 > 0$] controls the sequential localization, then say so directly and make it the central result.
2. The authors should show why this nonlinearity dominates here. On p. 11 you mention other SH-type reductions where a different nonlinearity drives localization. Why is this shell more sensitive to hoop effects than others? This comparison is important and currently underdeveloped.
3. The constitutive measurements on cut strips are not fully convincing. Cutting releases curvature and stress; bending a strip is not the same as bending the cylinder. If Eq. (5) is the crux, a stronger justification is needed to show that it represents the intact shell.
4. In Fig. 2B, four points with small error bars are consistent with the theory. Three additional points are well above the theoretical curve and have large error bars. Please explain this anomaly. Outliers? Different samples? In general, please explain how the error bars were calculated.
5. Are γ_1 and γ_2 generic to shells, or just features of worked aluminum cans? How general are the observations?
6. I don’t think that the paper demonstrates that hoop nonlinearity is the controlling effect. Perhaps show directly that changing γ_1 , γ_2 shifts the instability and post-buckling structure, while bending nonlinearity (Eq. 6) and global constraints (Boyle’s law, inextensibility) matter less.

Minor Comments

1. w_{\max} is never defined clearly. Is it constant between buckles, or does it vary?
2. Perhaps worth noting explicitly that the use of Boyle’s law is legitimate, since the compressibility comes from the trapped gas, not the water.
3. It is worthwhile to discuss the current results in the context of two previous works: (a) Calladine (2002), who offers an

alternative and relevant approach to initiation of an axisymmetric localized mode; and (b) Kreilos & Schneider (2017), who connect cylindrical shell buckling to the Swift–Hohenberg framework and snaking.

Reviewer #2

(Remarks to the Author)

I co-reviewed this manuscript with one of the reviewers who provided the listed reports. This is part of the Communications Physics initiative to facilitate training in peer review and to provide appropriate recognition for Early Career Researchers who co-review manuscripts.

Reviewer #3

(Remarks to the Author)

The manuscript explores a variation of a classical problem in shell mechanics. Specifically, it uses the generalised Swift-Hohenberg equation and a series of simplifying assumptions to numerically explore the buckling sequence in pressurised cylindrical metal cans. The writing is clear, and the results are presented to a good standard. There is a significant discrepancy between prediction and experiments, while some of the experimental procedures could have been presented with more detail. But the work is ambitious and has the potential to impact a wider range of similar buckling problems. This paper joins historical and recent research activity in this area and is likely to be of interest to a large number of mechanicians. More detailed comments, questions and suggestions are listed below:

1. What image processing technique was used to trace the profile in Fig. 1A? That information would be useful for reproducibility purposes but only Matlab is mentioned in this context without specifics.
2. In Fig. 2A, it might be useful to list the critical values for both experiment and prediction explicitly. It might also be useful to state how many repetitions of the experiment took place and how independence from strain rate was concluded - e.g. might be worth including Force-Strain plots for different strain rates.
3. The number of samples or repetitions is not stated clearly for any of the tests. How was repeatability established?
4. Material properties past yield would also depend on the rolling direction of the feedstock. Was that considered at all during the material characterisation tests?
5. The variation in can pressure and the uncertainty associated with that seem large in the experiments. At the same time, the theoretical model considers pressure, but the results do not examine its influence.
6. Reference [55] is not actually a citation - it's not clear why it was formatted like this.
7. The idea of using this compressive approach to intentionally create corrugations is interesting, but is not explored from a design or production engineering perspective. How would that be useful?
8. The discrepancy between the prediction and experiment for the critical loading is quite substantial, which might render the model of little practical use. However, one can see how more extensive physical experiments and better material characterisation might improve that. Some of the assumptions of the model might need to be amended as well - e.g. the shallow shell assumption is questionable for some of the geometries. The axial inextensibility assumption might also be restricting the fidelity of the model significantly, as the authors acknowledge.

Reviewer #4

(Remarks to the Author)

In this article, the authors uncover an original buckling pattern in partially liquid-filled cylindrical shells. The shells begin to yield via a single buckle that grows until it reaches a maximum height, triggering the formation of neighboring, equally spaced buckles. Once the regular wave-like pattern covers the entire shell surface and all buckles reach their maximum height, the shell bursts. The authors present compelling experimental evidence and derive an elegant model that captures essential features of the buckling instability. While the text, derivations, and figures clearly demonstrate how shell mechanics and geometry, coupled with non-linear material properties, lead to that instability, some discrepancies with the experimental results may benefit from being discussed more thoroughly.

I therefore recommend publication in "Communications Physics", given that the authors reply to the minor comments below.

- The model introduced by the authors successfully explains the phenomenology of the buckling instability and even recovers quantitative values for the buckling strains, forces, and wavelength. Yet, some fundamental discrepancies with the experiments are not addressed. For example, the maximum undulation height is ten times higher in the model; the wavelength remains independent of loading in the experiments but increases in the model; the initial localization in the model is accompanied by smaller satellite peaks, which are not observed in the experiments. The article would gain in interest and reach if these differences were acknowledged and potential explanations or directions for understanding them were discussed.

- In Fig. 2B, the authors show that the experimental and theoretical results can be well fitted by a linear relation between the average wavelength and the geometric mean of the shell's thickness and radius. However, this relation is not discussed further in the article, nor is the measured proportionality coefficient. Given the complexity of the governing equation (Eq. 7), deriving an analytical expression for the average wavelength is likely quite challenging. Nonetheless, could some simplifying assumptions enable the recovery of a known form of the Swift-Hohenberg equation? For instance, the authors show that the nonlinearities in the moment have limited importance. Ignoring this nonlinearity and setting the corresponding coefficient (α) to zero seems to simplify the equation considerably, bringing it closer to a more standard form.

- Three-point bending tests inherently show an inflection in the response due to geometrical non-linearities (for not small deflections). Have the authors verified that the measured inflection is not a geometrical non-linearity, but is indeed a material one? Also, the maximum curvature explored is $\kappa_x=0.035$ mm, or a radius of curvature of about 28 mm, which is of the order of the can radius but seems much larger than the radii of curvature in the buckles (by about a factor of 10, based on a quick estimate from the wavelength and w_{\max}).

- Is there a specific reason why the authors performed tensile stress tests along the hoop direction rather than the axial direction? Since the pattern emergence is driven by non-linearities in the tensile material, it would strengthen the message to verify that the constitutive relations for the forces in Eq. (17) are valid by measuring them independently.

Cosmetic comments and potential typos.

- In Fig. 1A-B, the vertical location of the buckles in the pictures does not match their locations in B. It would be more compelling to align them and show the entire vertical axis in B.

- To better appreciate the sensitivity to the nonlinear parameters (Fig. 3F-G, Fig. 8-9), the predictions of strains and forces could be shown as relative changes (%) from the averaged fitted values.

- In note [55] and just below Eq. (29), it is said that the equation is solved between 0 and 1, but while introducing z , it varies between $[-L/2, L/2]$. Shouldn't the integration interval then be $[-1, 1]$?

- In p.6, "the radius and the thickness a cylindrical shell" -> "the radius and the thickness of a cylindrical shell".

Reviewer #5

(Remarks to the Author)

This paper studies sequential, axisymmetric ring buckling in fluid-filled cans under axial compression.

The authors combine clean experiments with a 1-D model that reduces to a nonlinear Swift–Hohenberg–type equation. The model explains the ring-by-ring nucleation and growth seen in the tests. The paper is clear and interesting.

Please refer to the attached pdf file for detailed comments.

Version 1:

Reviewer comments:

Reviewer #1

(Remarks to the Author)

The revision is clearer and the narrative is more focused on hoop nonlinearity. The authors now suggest that the nonlinearity is somehow related to plasticity. The expanded discussion and additional modelling details are appreciated by the reviewer. However, central issues remains and I do not recommend publishing this work before they are resolved.

The extensional tests in Fig. S6 indicate that the material response remains essentially linear up to approximately 5 percent strain, with pronounced softening and subsequent stiffening emerging only between 5 and 10 percent. In the can experiments, the first buckle appears at axial engineering strains near 2 percent, and the deformations in figure 1B and in Fig. 3 suggest saturated buckle amplitudes corresponding to hoop strains on the order of 4 percent. This means that the nonlinearity hinged in Fig. S6 lies largely outside the strain range actually sampled by the shell during both the onset and the observed post-buckling evolution. The nonlinear is not what controls the initiation of buckling, and its role in the sequential localization remains unsubstantiated by strain measurements. Furthermore, the values of the pre-buckling strain calculated by the simulation (Fig 3B-3E) are well in excess of 10% which is both unreasonable and more than an order of magnitude larger than the measurements presented in Fig1B. This raises questions about the relevance of these strain results and the conclusions drawn from them. The authors would have to address this discrepancy and justify the validity of the simulation.

The status of the constitutive form in Eq. (5) also requires clearer framing. The polynomial law is described in the revision as Pseudo-plastic, yet this term does not correspond to a standard constitutive category, nor is a plasticity model implemented. The result is an empirical fit with clearly visible bias relative to the data, but the the SH framework and corresponding buckling mechanism depends critically on the specific signs of this polynomial. Therefore, this may not be a general mechanism, but a forced fit of the framework to the data. A demonstration that a valid plasticity law, over the relevant strain range, would yield the same qualitative structure would substantially strengthen the claim. At a minimum, the authors should include a clear statement in the main text that Eq. (5) is an approximate empirical representation chosen to obtain a tractable reduced model, which would help prevent overinterpretation of the constitutive mechanism. Clarifying these issues is critical for a publishable coherent picture.

Reviewer #3

(Remarks to the Author)

The authors provided a thorough response to earlier comments and produced satisfactory amendments to the manuscript, making it suitable for publication.

The only exception is point 4 in the rebuttal letter, which I believe misinterprets my reference to the anisotropy in the feedstock. Anisotropy is almost universally present in sheet metal because sheet metal is produced by the rolling process, which has inherent directionality. It may very well be that the sheet metal anisotropy has little influence over the phenomenon the paper is examining, but it certainly has an impact on the forming operation, resulting in the well-known phenomenon of "earing".

Reviewer #4

(Remarks to the Author)

The authors have provided clear and convincing responses to my comments and implemented relevant revisions.

I have also looked through all the changes and the other reviewers' comments.

In my opinion, the authors satisfyingly addressed the questions and concerns raised by the reviewers. I would therefore recommend publication of the manuscript in its current form.

Reviewer #5

(Remarks to the Author)

This is an elegant and well-executed study of sequential, axisymmetric buckling in fluid-filled cylindrical shells. The combination of simple, convincing experiments and a minimal nonlinear Swift–Hohenberg–type model provides new physical insight into sequential localization and homoclinic snaking in realistic materials.

The revision has fully addressed prior comments:

- The discussion of manufacturing feasibility has been appropriately qualified.
- The distinction from prior Yoshimura-type (diamond) buckling work is now explicit.
- The wavelength scaling and fit quality are clarified and consistent.
- The irreversibility of deformation on unloading is clearly stated.
- The limitations of the inextensibility assumption and boundary effects are acknowledged.
- Parameter sensitivity is extended and well-documented in the Supplementary Information.

The paper is now clear, balanced, and reproducible. I recommend acceptance after minor editorial polishing (e.g., typographic consistency and equation spacing).

This work will likely stimulate further studies of pattern formation driven by material nonlinearities in shell mechanics and soft matter.

Version 2:

Reviewer comments:

Reviewer #1

(Remarks to the Author)

The revision is more transparent about the modelling assumptions and limitations. Some of my concerns remain, in particular the phenomenological nature of the constitutive law and the unrealistically large strains reached in the simulations; thus, in my view the modelling should be viewed as illustrative rather than predictive. Within that framing, the work presents a nice experimental observation supported by a theoretical framework.

I. REVIEWER 1

We extend our gratitude to the reviewer for taking the time to read the manuscript and for providing constructive feedback. An itemised response to their comments is given below.

General feedback:

The paper is clear and carefully executed, but in its present form it does not sufficiently emphasize what is the new physics. The progression of sequential buckles in soda cans is well known.

We have significantly expanded our discussion (pages 12 and 13) to address this point. As discussed in § V, and also introduced in the second paragraph of the Introduction, our study is original in exploring buckling of liquid-filled cylindrical shells. In contrast to all other studies of sequential buckling, our system develops *ring buckles* that preserve axi-symmetry of the system. Moreover, we have developed an original model, underpinned by experimental observations and empirical measurements, that explains the new phenomenon as “a consequence of nonlinear hoop stresses that soften and re-stiffen in response to axial compression and internal pressurization”. This mechanism of buckling is unique as stressed in the discussion on page 12. Furthermore, our work is a rare attempt to combine *experimental* realizations of sequential buckling with a theoretical model, as also stated explicitly in the Introduction to the paper.

... the \sqrt{Rt} scaling of the wavelength is trivial from dimensional arguments.

The final shape of surface buckles in our filled cans is obtained via localisation, rather than being set by a supercritical transition to wrinkling, as studied in the classical literature from the 60s [1]. Furthermore, for systems with finite domains, unlike the ones investigated classically, localised patterns can also display length-scales depending on the domain length. Indeed, [2] is an example where the nature of the spacing between the localisation phenomena does not hold for all parameter values. Thus, the fact that a similar scaling holds is a neat coincidence rather than anything that should be anticipated. Note that Reviewer 4 agrees with us, stating “Given the complexity of the governing equation (Eq. (7)), deriving an analytical expression for the average wavelength is likely quite challenging”. In response to their comment, we have now expanded the paragraph on the \sqrt{Rt} scaling on page 6 of the manuscript.

[1] J. Hutchinson, Axial buckling of pressurized imperfect cylindrical shells, AIAA J. 3, 1461 (1965)

[2] Z.D. Li et al. Traveling spatially localized convective structures in an inclined porous medium, Phys. Rev. Fluids 10, 034402 (2025).

The reduced model reproduces the experimental data, as it is reverse engineered to do so: The authors fit nonlinearities from cut-strip tests, insert them into a Swift–Hohenberg-like balance, impose the correct global constraints (Boyle’s law for the headspace, axial inextensibility), and unsurprisingly obtain the observed localization and serrated force–strain curves.

The model is derived from the well-established DMV shell equations, and the arrival at a Swift-Hohenberg-like (SH-like) equation was a consequence of empirical input rather than “reverse engineering”. Our methodology is typical of any theoretical modelling: using well-established equations, we made physical assumptions and combined them with empirical experiments to yield predictions that have been compared against an *independent* set of experiments. The derivation is argued carefully in the main manuscript and the Supplementary Information which outlines all the steps taken. Note that we could have achieved better quantitative agreement with experiments by performing parameter fitting to the can buckling experiment to obtain γ_1 and γ_2 as opposed to conducting independent tests on the strip samples (see the response to the first comment of Reviewer 4 and the discussion on page 12). However, that would have been rightly construed as “reverse engineering”. We have acknowledged the limitations of our model, and have expanded our manuscript significantly in response to the reviewers’ comments by presenting more details of the modelling.

I understand correctly, the real novelty lies in Eq. (5), with the fitted parameters $\gamma_1 < 0$ and $\gamma_2 > 0$. If this is the case, then the article will need revision to convincingly stress that as the main result.

As argued above, there are many novel aspects of our work, which include but are not limited to experimentally investigating sequential (ring) buckling, empirically incorporating the nonlinear material properties (including the hoop-stresses), and proposing a mechanism that underlies the observed sequentialised phenomena. However, the reviewer is right in their observation that having $\gamma_1 < 0$ and $\gamma_2 > 0$ is crucial for the buckling mechanism that we describe. In the concluding sentences of the Introduction, we already stressed that “the sequential emergence of ring buckles is a consequence of nonlinear hoop stresses that soften and re-stiffen in response to axial compression and internal pressurization”. We emphasize this point in all sections of the manuscript, except §2 where we discuss the buckling phenomenon for the first time. Our changes to the manuscript in response to the queries below emphasize

and evidence the novel mechanism of buckling further.

Specific feedback:

1. *The main point of the paper is not made clearly. If the real physics is that the nonlinear hoop law [Eq. (5), with $\gamma_1 < 0$ and $\gamma_2 > 0$] controls the sequential localization, then say so directly and make it the central result.*

We have now expanded the discussion of the nonlinear constitutive response on page 11 of the manuscript, thereby stressing the importance of the hoop stresses further. We point out that this has already been covered in all sections of the paper, except §2 where the experiment was introduced.

2. *The authors should show why this nonlinearity dominates here. On p. 11 you mention other SH-type reductions where a different nonlinearity drives localization. Why is this shell more sensitive to hoop effects than others? This comparison is important and currently underdeveloped.*

We have significantly expanded our discussion on page 12, explaining the difference between our results and that of other works where SH-type models feature. Simply put, the mechanisms covered in the literature so far are not compatible with the axi-symmetry of patterns that we observe in the formation of ring buckles. We quote the relevant section here:

“Similarly, while previous work [16, 33] examined the localisation of diamond-shaped buckles on cylindrical shells, and proposed an underlying SH-type mechanism for the destabilisation of the shells to dimple-like perturbations, the nonlinearities required for their analogy to the SH equations came from the nonlinear force response of the shell to point indentation, which is based on experiments described in [15]. In contrast, we demonstrate that multiplicity of buckled states can emerge from a competition between softening and re-stiffening nonlinearities of the shell material, permitting localized solutions to emerge sequentially via homoclinic snaking, even as rotational symmetry is preserved.”

3. *The constitutive measurements on cut strips are not fully convincing. Cutting releases curvature and stress; bending a strip is not the same as bending the cylinder. If Eq. (5) is the crux, a stronger justification is needed to show that it represents the intact shell.*

While cutting a strip does release curvature as the reviewer points out, the tests that we performed still provide first-order estimates of the relevant material properties. In particular, in order to perform the extensional tests used to determine the hoop-stress stiffness, first we had to flatten the cut-out section. Using a three-point bend test, we measured the moment-curvature response of a circumferential section of the can during flattening, with initial length of the strip of exactly half the circumference. Note that the curvature of the semi-circular strip at the beginning of our test was $\sim 1/26.5 \text{ mm}^{-1}$. Hence, cutting of the semi-circular section did not change the curvature significantly.

Throughout flattening, the moment-curvature relation was linear, see Fig. 1 below, and included forces that were at least a few orders of magnitude smaller than the forces associated with the extensional tests. This suggests that flattening should not significantly influence the extensional tests performed to assess the hoop-stress stiffness. We would also like to point to [3] for previous work that lists various testing methods for hoop-stresses of metallic tubes, the simplest of which resembles our tests in Part B of the Supplementary Information.

Finally, it is technically challenging to test the bending properties of the entire can, particularly when it is filled with liquid. While we agree that its bending behaviour may differ from that of the cut strips, we anticipate that localized bending in the vicinity of the ring buckles will closely resemble the material bending properties measured using strips. Hence, we argue that testing of the bending properties of the cut strips is a suitable alternative to finding the material response of the whole can. We have followed the reviewer’s suggestion by amending our Supplementary Information on pages 22 and 23 of the combined document to briefly discuss these points.

[3] Dick, Chris P., and Yannis P. Korkolis. ”Mechanics and full-field deformation study of the ring hoop tension test.” *International Journal of Solids and Structures* 51.18 (2014): 3042-3057.

4. *In Fig. 2B, four points with small error bars are consistent with the theory. Three additional points are well above the theoretical curve and have large error bars. Please explain this anomaly. Outliers? Different samples? In general, please explain how the error bars were calculated.*

In our experiments, we use commercially available beverage cans, so we have tested the shell geometries that we could procure ‘off-the-shelf’ from convenience stores. Some patterns were more regular than others, which

FIG. 1. **Moment-curvature relation of circumferential bending test of the cans.** The recorded moments against the curvatures at the point of force application during the three-point bend test of a semi-circular section of the can circumference. The two tested samples were of width $W = 4$ mm, and were supported at the points shown with dots that we separated by the distance of $L = 13.25$ mm.

affected the size of our error bars. We have expanded the Supplementary Information by including two additional section on experiments, see pages 31 and 32 of the combined document, which also describe how the wavelengths were measured.

5. *Are γ_1 and γ_2 generic to shells, or just features of worked aluminum cans? How general are the observations?*
The signs of γ_1 and γ_2 are universal to shells made of a wide range of materials that soften after yielding, but then re-stiffen under increasing stress. Consequently, our observations are likely to be general to any shells made of materials of this type. Indeed, we have observed the same qualitative behaviour in cans made of steel. As such, we believe that our results stand for shells made of any materials that soften after yielding, but then re-stiffen under increasing stress. We have changed the first paragraph in the Discussion section on page 12 to emphasize this point.

6. *I don't think that the paper demonstrates that hoop nonlinearity is the controlling effect. Perhaps show directly that changing γ_1, γ_2 shifts the instability and post-buckling structure, while bending nonlinearity (Eq. 6) and global constraints (Boyle's law, inextensibility) matter less.*

We have already varied α , i.e., the bending nonlinearity, demonstrating its limited effect on the system. Although this is discussed in the main text of the manuscript, the details of the calculation are in the Supplementary Information because we wanted to avoid describing the same (potentially tedious) technical steps in the manuscript. Following the reviewer's suggestion, we now also include a sentence on page 11 of the manuscript on the influence of the initial value of the internal pressure, and elaborate on it in the Supplementary Information on pages 29-31 of the combined document. The inextensibility condition is discussed elsewhere, see pages 8, 10, 12 and 25 of the combined document.

Minor comments

1. *w_{max} is never defined clearly. Is it constant between buckles, or does it vary?*

The quantity w_{max}/R has already been defined in the caption to Fig. 1, and its numerical value given in Figs. 1 and 3 for the particular experiments/numerics. We have now expanded the caption to Fig. 1 to clarify that w_{max}/R is consistent in experiments with the same geometry, but varies between geometries. Furthermore,

following the reviewer's comment, we have measured w_{max} in our experimental samples and have included this data in Table II of the Supplementary Information on page 32 of the combined document.

2. *Perhaps worth noting explicitly that the use of Boyle's law is legitimate, since the compressibility comes from the trapped gas, not the water.*

This is now explicitly stated on page 8 of the manuscript.

3. *It is worthwhile to discuss the current results in the context of two previous works: (a) Calladine (2002), who offers an alternative and relevant approach to initiation of an axisymmetric localized mode; and (b) Kreilos & Schneider (2017), who connect cylindrical shell buckling to the Swift-Hohenberg framework and snaking.*

On page 12 of the manuscript, we now elaborate on our comparison with Kreilos and Schneider (2017) further. We have also added a comparison with Calladine (2002) to our discussion on page 13 of the manuscript.

II. REVIEWER 3

We extend our gratitude to the reviewer for taking the time to read the manuscript and for providing constructive feedback. We are glad to hear that they found our work “ambitious” and having wide impact potential. An itemized response to the reviewer’s comments is given below:

1. *What image processing technique was used to trace the profile in Fig. 1A? That information would be useful for reproducibility purposes but only Matlab is mentioned in this context without specifics.*

We have amended the Supplementary Information for our manuscript to add a description of our image processing techniques, see page 31 of the combined document.

2. *In Fig. 2A, it might be useful to list the critical values for both experiment and prediction explicitly. It might also be useful to state how many repetitions of the experiment took place and how independence from strain rate was concluded - e.g. might be worth including force-strain plots for different strain rates.*

We have updated the caption to Fig. 2 to display the critical values obtained in the experiment and using the model. We have expanded the Supplementary Information by including a table with the number of tested samples per geometry used in averaging (see page 32 of the combined document). The grey lines in Fig. 2A already correspond to data obtained by varying compression speeds in the range 0.1 – 8.5 mm/s, as pointed out in the caption. We have now also created a Supplementary Video to show the buckling phenomenon in liquid-filled cans deformed at different compression rates.

3. *The number of samples or repetitions is not stated clearly for any of the tests. How was repeatability established?*

We have amended the discussion of our material tests on pages 21 and 23 of the combined document to explicitly state how many test were carried out for each type of test. As explained above, we have also expanded the Supplementary Information by including a table with the number of tested samples per geometry.

4. *Material properties past yield would also depend on the rolling direction of the feedstock. Was that considered at all during the material characterisation tests?*

Aluminium beverage cans are formed by punching out a cup-form into the final shape of the can. This was taken into account by testing the bending response in the axial direction (parallel to the direction of the punch) and the stretching response in the circumferential direction (perpendicular to the direction of the punch). We assume that the other deformation modes (axial stretching and bending in the circumferential direction), as well as any mixed modes are negligible in the system. We now explicitly discuss how the can manufacturing affects modelling assumptions in the Supplementary Information, on page 25 of the combined document.

5. *The variation in can pressure and the uncertainty associated with that seem large in the experiments. At the same time, the theoretical model considers pressure, but the results do not examine its influence.*

Experiments with pressurised cans and with (unpressurized) cans filled at atmospheric pressure resulted in the same phenomena. So despite significant variations in the initial internal pressure of cans, we believe that internal pressure does not have a significant effect on the mechanics of the system, at least in the regime discussed by this manuscript. We varied the initial pressure Q_i in our model, and have added a short discussion of the effect this has in the Supplementary Information on pages 29-31 of the combined manuscript. We have also amended the text on pages 6 and 11 of the manuscript to clarify this point.

6. *Reference [55] is not actually a citation - it’s not clear why it was formatted like this.*

We have changed this to include the footnote directly in the text, on page 27 of the combined document.

7. *The idea of using this compressive approach to intentionally create corrugations is interesting, but is not explored from a design or production engineering perspective. How would that be useful?*

We have expanded our discussion of this topic on page 13 of the manuscript by adding a clarifying sentence.

8. *The discrepancy between the prediction and experiment for the critical loading is quite substantial, which might render the model of little practical use. However, one can see how more extensive physical experiments and better material characterisation might improve that. Some of the assumptions of the model might need to be amended as well - e.g. the shallow shell assumption is questionable for some of the geometries. The axial inextensibility assumption might also be restricting the fidelity of the model significantly, as the authors acknowledge*

We believe that the primary reason for the discrepancy in the critical loading and strain is the complex pre-buckling behaviour of the cans observed in the experiments. The current model, which links the force and the strain via the in-extensibility constraint, cannot reproduce the nonlinear response seen for small strains, $\varepsilon < \varepsilon_{cr}$, as shown in Fig. 2A. We have expanded page 10 of the manuscript to discuss this point.

III. REVIEWER 4

We extend our gratitude to the reviewer for taking the time to read the manuscript and for providing constructive feedback. We are glad that they found our work “original”, and praised “compelling experiments” and “elegant modelling”. An itemized response to the reviewer’s comments is given below:

1. *The model introduced by the authors successfully explains the phenomenology of the buckling instability and even recovers quantitative values for the buckling strains, forces, and wavelength. Yet, some fundamental discrepancies with the experiments are not addressed. For example, the maximum undulation height is ten times higher in the model; the wavelength remains independent of loading in the experiments but increases in the model; the initial localization in the model is accompanied by smaller satellite peaks, which are not observed in the experiments. The article would gain in interest and reach if these differences were acknowledged and potential explanations or directions for understanding them were discussed.*

We have added further discussion of results, on pages 10 and 12 of the manuscript: On page 10, we have added an explanation of the discrepancy in the critical force and the strain at the onset of the instability. As pointed out in the manuscript already on page 12, the discrepancy in the saturation amplitude can be presumably removed by tuning the values of γ_1 and γ_2 : our parameter exploration shows that changing the values of these coefficients moves the amplitudes of theoretically predicted undulations in the right direction, towards greater agreement with experiments. However, for our study we have instead used the values determined independently from complementary experiments and we stand by that choice.

The buckling wavelength predicted by our model does not change significantly with increases in load i.e., as more buckles appear. We verified this by measuring the average peak-to-peak separation in the can surface profiles as a function of engineering strain, as shown in Fig. 2 below. Overall, the wavelength changes by less than 1% in Fig. 2, and we link this to the boundary effects in our model. We believe that the inferred increase in wavelength that the reviewer refers to is a consequence of presenting our results in the coordinate system of the can, given by $x/(L - \delta x)$, such that it appears that the wavelength changes as δx changes. This is now clarified on page 10 of the manuscript.

The observed sine-like patterns have tanh-like modulation which is quite common in solutions to SH equations [4]. There appears to be a lengthscale set by the modulation that determines the prominence of the satellite peaks. However, we have not investigated this further, and have now explicitly acknowledged the limitation of our study on page 12.

[4] Hunt, Giles W., et al. ”Cellular buckling in long structures.” *Nonlinear Dynamics* 21.1 (2000): 3-29.

FIG. 2. **Wavelength variations with strain.** The wavelength predicted numerically (and measured by averaging the peak-peak distance in the surface profile) for varying engineering strain. The difference in the predicted wavelengths is less than 1%.

2. *In Fig. 2B, the authors show that the experimental and theoretical results can be well fitted by a linear relation between the average wavelength and the geometric mean of the shell’s thickness and radius. However, this relation is not discussed further in the article, nor is the measured proportionality coefficient. Given the complexity of the governing equation (Eq. 7), deriving an analytical expression for the average wavelength is likely*

quite challenging. Nonetheless, could some simplifying assumptions enable the recovery of a known form of the Swift-Hohenberg equation? For instance, the authors show that the nonlinearities in the moment have limited importance. Ignoring this nonlinearity and setting the corresponding coefficient (α) to zero seems to simplify the equation considerably, bringing it closer to a more standard form.

As the reviewer points out, the final governing equation of our system is quite complex, even if the value of α is set to zero, and requires numerical solution. However, setting $\gamma_1 = \gamma_2 = 0$ reduces the model further to the one studied in the classical literature on shell buckling. For example, [1] has shown that an average wavelength of (globally occurring) axisymmetric buckling is expected to follow $\lambda = \varepsilon C \sqrt{Rt}$ in case of linear elastic shells with internal pressure. Here, C is a constant ≈ 3.479 for a Poisson ratio of $\nu = 0.35$, and ε is an integer. Remarkably, for ε of 2, this scaling is close to our experimental observations. Thus, despite all the complexities of our phenomena, the experimentally measured wavelength is remarkably consistent with the classical result. We have amended the text on page 6 of the manuscript to include this observation.

[1] Hutchinson, J. "Axial buckling of pressurized imperfect cylindrical shells", AIAA journal, 3(8), 1461-1466.

3. *Three-point bending tests inherently show an inflection in the response due to geometrical non-linearities (for not small deflections). Have the authors verified that the measured inflection is not a geometrical non-linearity, but is indeed a material one? Also, the maximum curvature explored is $\kappa_x = 0.035$ mm, or a radius of curvature of about 28 mm, which is of the order of the can radius but seems much larger than the radii of curvature in the buckles (by about a factor of 10, based on a quick estimate from the wavelength and w_{max}).*

During the three-point bending test, we permanently deformed our samples. This was obvious from observing our samples when they were unloaded, and we therefore believe that the nonlinearities modelled by the equation (6) (which corresponds to the equation (18) in the Supplementary Information) arise primarily from yielding of the metal. We now clarify the point by including a sentence in the Supplementary Information on page 22 of the combined document. As pointed out by the reviewer, ideally we would like to deform our samples further to see the effect of geometric non-linearities. However, with the current setup we are unable to measure our bending moments at much higher values of the curvature - the data is technically challenging to obtain because the samples slip on the supports in the three-point bending test, for example. Considering that we have already observed material non-linearities at the smaller curvatures that were examined, as well as the fact that the bending non-linearities are, according to our model, of limited importance, we believe it is unnecessary to explore geometric nonlinearities at higher curvatures.

4. *Is there a specific reason why the authors performed tensile stress tests along the hoop direction rather than the axial direction? Since the pattern emergence is driven by non-linearities in the tensile material, it would strengthen the message to verify that the constitutive relations for the forces in Eq. (17) are valid by measuring them independently.*

We have expanded the discussion of these points in the Supplementary Information on pages 20 and 25 of the combined document. We have measured the circumferential stretching stiffness and the axial bending stiffness only, as the assumptions of axi-symmetry and in-extensibility in the axial direction preclude any axial stretching or circumferential bending. There should definitely be further investigation of the role of the mixed deformation modes and understanding the effect of relaxing the 'no axial stretching' assumption, but this is beyond the scope of the current study.

Minor comments

1. *In Fig. 1A-B, the vertical location of the buckles in the pictures does not match their locations in B. It would be more compelling to align them and show the entire vertical axis in B.*

We have significantly expanded the Supplementary Information to include the discussion of boundary effects/conditions (see page 25 of the combined document), and have elaborated on page 10 regarding the numerical data in Fig. 2B. On page 11, we now also discuss the possibility of seeing the simultaneous appearance of ring buckles near the middle of a can and at its boundary under compression. While this occurs quite rarely, the (nicely illuminated) sample that we chose for the montage in Fig. 1 happens to have developed such buckles. In order to avoid confusing readers at the start of the manuscript, we have truncated our space-time plot in Fig. 1B to only show the main sequential localization phenomenon.

2. *To better appreciate the sensitivity to the nonlinear parameters (Fig. 3F-G, Fig. 8-9), the predictions of strains and forces could be shown as relative changes (%) from the averaged fitted values.*

We have opted to leave the data as they are in the figures for easier direct comparison with experiments.

3. *In note [55] and just below Eq. (29), it is said that the equation is solved between 0 and 1, but while introducing z , it varies between $[-L/2, L/2]$. Shouldn't the integration interval then be $[-1, 1]$?*
As explained in the manuscript on page 10, we have performed parameter continuation in Auto-07p, by solving our model on the half-domain and assuming symmetry across the centreline. We have clarified this again in the Supplementary Information on page 28 of the combined document.
4. *In p.6, "the radius and the thickness a cylindrical shell" \rightarrow "the radius and the thickness of a cylindrical shell".*
The typo has been fixed.

IV. REVIEWER 5

We extend our gratitude to the reviewer for taking the time to read the manuscript and for providing constructive feedback. We are glad that they are supportive of our work. An itemized response to the reviewer's comments is given below:

1. ***Tone down the manufacturing claim.***

Please soften language on “feasibility of post-fill corrugation / streamlining production,” or move it to Outlook with caveats (repeatability, burst margin, QC, end constraints). See Abstract last lines and Discussion last paragraph.

As the reviewer notes, the idea of using localised buckling in filled cylindrical cans for cold-forming, while an exciting prospect, requires a lot of further investigation which is beyond the scope of our study. Thus, we have accepted the reviewer's suggestions, and have dropped the manufacturing claim from the abstract. Instead, we state some of the caveats of our idea in the outlook of the paper on page 13 of the manuscript.

2. ***Make the contribution vs prior “diamond/Yoshimura” work extra explicit.***

In the Introduction/Discussion, add 2–3 sentences: this work = rings + material non-linearities; prior = diamonds + indentation-driven non-linearity. Cite and contrast clearly.

We have expanded the discussion on page 12 of the manuscript to directly contrast our work with the prior work on localisation of the diamond buckles.

3. ***Fix the wavelength prefactor mismatch and report fit quality Fig. 2B text/legend list $A = 6.74 \pm 1.78$ and $A = 6.47 \pm 1.78$. Please make these consistent and report CI/R^2 in the caption or main text.***

The typo has been fixed and we are now citing the R-squared value in the caption to Fig. 2.

4. ***State what happens on unloading (hysteresis or residual corrugation). You say results are independent of strain/compression rate. Add one line on unloading: was hysteresis tested? any permanent corrugation? If not tested, please say so. Align wording across Results and Methods.***

In our experiments, shown in Fig. 1, we have studied the buckling phenomena when the cans are loaded. The resulting surface buckles do not measurably change when the cans are unloaded, which suggests that their surface deforms plastically. However, since we are interested in the loading phase of the experiment, we have taken a pseudo-plastic approach in our model by incorporating empirically determined material nonlinearities. This is now discussed explicitly in our manuscript on page 5 and page 6.

5. ***Model closures: show a small sensitivity.***

a) Axial inextensibility. *After Eq. (9)/(25), acknowledge this strong assumption and, if possible, include a tiny test with small axial stretch to see its effect on w_{max} and $\varepsilon_{max}(SI)$.*

As pointed out by the reviewer, the axial in-extensibility condition is a very strong constraint on the system. This, combined with our assumption of axisymmetry implies that the cans do not experience circumferential bending or axial stretching. Hence, the anisotropy is a prominent feature of our model, and is further captured by the different functional forms of the hoop stresses and the axial bending. We now stress this more explicitly in the Supplementary Information on page 25 of the combined document.

Considering our experiments were conducted with commercially available beverage cans, it is quite difficult for us repeat any experiments by introducing axial pre-stretching first. This would require substantial changes in our setup, or the samples studied. While the results of such tests would be very interesting, and build upon the current study, we believe that our main findings can be allowed to stand on their own.

6. ***Boundary conditions and end effects.*** *You apply simply supported ends in the model, but real cans have domed ends and friction; first-buckle location varies between tests. Add one sentence on this limitation and its likely bias on nucleation.*

We have expanded the Supplementary Information on page 25 of the combined document to discuss this further.

7. ***Robustness to constitutive parameters (compact).*** *You vary γ_2 (nice). Consider adding a small (γ_1, γ_2) “phase sketch” (even schematic) showing where sequential localized branches exist. This can go to SI near the*

current sensitivity plots.

In the Supplementary Information, on page 30 of the combined document, we have added a schematic showing the parameter values that were explored in our paper in detail. However, we expect that the localised solutions exist outside the parameter bounds studied in this manuscript. There is plenty of evidence that supports this conjunction, albeit in simpler systems (see, e.g., Ref. [30]). We limit ourselves to the parameter range of relevance to our experiments.

Minor comments

1. *Check notation/units are consistent (e.g., define Q_L , keep symbols uniform)*

This has been checked. Note that Q_L has already been defined where it was used for the first time, on page 7 of the manuscript.

2. *In Fig. 3, consider tiny cartoons next to B–E and label $w_{max}, \varepsilon_{max}$ once on the panel for readers.*

Fig. 3 is very dense and contains a lot of labels already. Hence, we have supplemented the figure with a detailed caption to avoid overloading it. For example, in the caption we explain that the dashed line in Fig. 3E corresponds to w_{max} . Note that we have tried the reviewer's suggestions and have found the figure to be harder to read.

I. REVIEWER 1

We extend our gratitude to the reviewer for taking the time to read the manuscript and for providing constructive feedback. An itemised response to their comments is given below.

The extensional tests in Fig. S6 indicate that the material response remains essentially linear up to approximately 5 percent strain, with pronounced softening and subsequent stiffening emerging only between 5 and 10 percent. In the can experiments, the first buckle appears at axial engineering strains near 2 percent, and the deformations in figure 1B and in Fig. 3 suggest saturated buckle amplitudes corresponding to hoop strains on the order of 4 percent. This means that the nonlinearity highlighted in Fig. S6 lies largely outside the strain range actually sampled by the shell during both the onset and the observed post-buckling evolution.

First, we emphasize the distinction between the global engineering strain of the can shown in Figs. 1-3, and previously defined as $\varepsilon = \delta x/L$, and the local engineering strains on the circumference that are measured in the extensional tests, $\varepsilon = \delta^{str}/\mathcal{L}^{str}$, which is shown in Fig. S6 and which correspond to the hoop-stresses. We appreciate the confusion caused by using the same notation, ε , for these distinct strains in Figs. 1-3 and Fig. 6S, although the definition of the reported strain was different in the two cases, as explained in Section B of the Supplementary Material where we introduced the local strain. For clarity, we have now changed the notation for the local strain measured in the extensional tests to $\varepsilon^{str} = \delta^{str}/\mathcal{L}^{str}$ (see Fig. S6). Furthermore, we agree that the global engineering strain associated with the first buckle appearing is $\sim 2\%$ as the reviewer notes. However, we highlight that the hoop strain in the buckle ($\approx w_{max}/R$) is $> 4\%$, which is only marginally different from the $\sim 5\%$ range quoted by the reviewer and, as such, the relevant nonlinearity is present within the strain range experienced by the buckle.

Furthermore, the values of the pre-buckling strain calculated by the simulation (Fig. 3B-3E) are well in excess of 10% which is both unreasonable and more than an order of magnitude larger than the measurements presented in Fig1B. This raises questions about the relevance of these strain results and the conclusions drawn from them. The authors would have to address this discrepancy and justify the validity of the simulation.

We highlight that, at onset, the predicted strains are both reasonable and in good agreement with the experimental results. To clarify, the experimentally measured ε_{cr}^{exp} relate to both the point when the force-strain curves develop the first localised peak, as shown in Fig. 2, and the strain at which the localised solution branches of an imperfect bifurcation arise. The critical strain obtained numerically, ε_{cr}^{pred} , is defined in the same way, as is now explicitly stated in the caption to Fig. 3 (previously it was only illustrated in Fig. 3A). Indeed, we find that $\varepsilon_{cr}^{pred} = 0.0283$, which is in good quantitative agreement with the measured $\varepsilon_{cr}^{exp} = 0.021$.

Beyond onset, the quantitative agreement fares less well. However, we already acknowledge the quantitative difference between our simulation results and experimental measurements beyond onset, and even propose the mechanisms for these discrepancies in our discussion, quoted here for reference:

“Although the model qualitatively captures localization of the axisymmetric buckles, the predicted maximum amplitude of the buckles is significantly larger than its experimentally measured value, which inevitably leads to a significant difference in the maximum strains as well, via the assumption of inextensibility. It is possible to tune the strength of nonlinearities in our model, via the values of γ_1 and γ_2 , to improve quantitative agreement between its predictions and the experimental results. However, the real strength of the model is in revealing the underlying mechanism for sequential buckling in a cylindrical shell.”

The status of the constitutive form in Eq. (5) also requires clearer framing. The polynomial law is described in the revision as Pseudo-plastic, yet this term does not correspond to a standard constitutive category, nor is a plasticity model implemented. The result is an empirical fit with clearly visible bias relative to the data, but the the SH framework and corresponding buckling mechanism depends critically on the specific signs of this polynomial. Therefore, this may not be a general mechanism, but a forced fit of the framework to the data. A demonstration that a valid plasticity law, over the relevant strain range, would yield the same qualitative structure would substantially strengthen the claim. At a minimum, the authors should include a clear statement in the main text that Eq. (5) is an approximate empirical representation chosen to obtain a tractable reduced model, which would help prevent overinterpretation of the constitutive mechanism.

We have now added a clarification on page 8 in the main text to stress the empirical nature of our modelling approach:

“We stress that the polynomial relationships (5) and (6), and hence the values of these coefficients, are empirical and

have been chosen to obtain a tractable reduced model.

We would also like to highlight that in analysis of Winkler beams on non-linear foundations, for example, adding 2nd and 3rd order nonlinear terms to the elastic response of the foundation is an established technique for capturing the non-trivial behaviour of e.g., soil foundations [1,2]. Perhaps more pertinently, we have previously captured the post-buckling behaviour of plastically deforming columns [3] by adding higher-order terms to the elastic response of the columns. These terms were fit empirically to micromechanical measurements and rendered a simplified model of the problem with quantitative predictive power. Indeed, it was a synthesis of these approaches that prompted our empirical approach to modelling the constitutive nonlinearities with polynomials.

[1] Yankelevsky, D.Z., Eisenberger, M. and Adin, M.A., 1989. Analysis of beams on nonlinear winkler foundation. *Computers & Structures*, 31(2), pp.287-292.

[2] Tsiatas, G.C., 2010. Nonlinear analysis of non-uniform beams on nonlinear elastic foundation. *Acta Mechanica*, 209(1), pp.141-152.

[3] Jain, S., Box, F., Johnson, C. and Pihler-Puzović, D., 2023. Material nonlinearities yield doubly negative holey metamaterials. *Extreme Mechanics Letters*, 64, p.102065.

II. REVIEWER 3

The authors provided a thorough response to earlier comments and produced satisfactory amendments to the manuscript, making it suitable for publication. The only exception is point 4 in the rebuttal letter, which I believe misinterprets my reference to the anisotropy in the feedstock. Anisotropy is almost universally present in sheet metal because sheet metal is produced by the rolling process, which has inherent directionality. It may very well be that the sheet metal anisotropy has little influence over the phenomenon the paper is examining, but it certainly has an impact on the forming operation, resulting in the well-known phenomenon of "earing".

We thank the reviewer for their feedback and support for the manuscript. We appreciate the clarification they have offered regarding point 4, and agree with them that the material would be inherently anisotropic due to the rolling process. However, this does not seem to impact the results of our investigation.

III. REVIEWER 4

The authors have provided clear and convincing responses to my comments and implemented relevant revisions. I have also looked through all the changes and the other reviewers' comments. In my opinion, the authors satisfactorily addressed the questions and concerns raised by the reviewers. I would therefore recommend publication of the manuscript in its current form.

We thank the reviewer for their feedback and support of the manuscript.

IV. REVIEWER 5

This is an elegant and well-executed study of sequential, axisymmetric buckling in fluid-filled cylindrical shells. The combination of simple, convincing experiments and a minimal nonlinear Swift–Hohenberg-type model provides new physical insight into sequential localization and homoclinic snaking in realistic materials. The revision has fully addressed prior comments:

- *The discussion of manufacturing feasibility has been appropriately qualified.*
- *The distinction from prior Yoshimura-type (diamond) buckling work is now explicit.*

- *The wavelength scaling and fit quality are clarified and consistent.*
- *The irreversibility of deformation on unloading is clearly stated.*
- *The limitations of the inextensibility assumption and boundary effects are acknowledged.*
- *Parameter sensitivity is extended and well-documented in the Supplementary Information.*

The paper is now clear, balanced, and reproducible. I recommend acceptance after minor editorial polishing (e.g., typographic consistency and equation spacing). This work will likely stimulate further studies of pattern formation driven by material nonlinearities in shell mechanics and soft matter.

We thank the reviewer for their feedback and support of the manuscript.

Summary

This paper studies **sequential, axisymmetric ring buckling** in fluid-filled cans under axial compression. The authors combine clean experiments with a **1-D model** that reduces to a **nonlinear Swift–Hohenberg–type equation**. The model explains the ring-by-ring nucleation and growth seen in the tests. The paper is clear and interesting.

Main strengths

- Simple, convincing experiments with force–strain curves and profiles.
- Careful material characterization; anisotropy and nonlinearities are measured and used.
- Clear link to **localized states / homoclinic snaking** and pattern wavelength scaling $\langle \lambda \rangle \propto \sqrt{Rt}$

What to improve (actionable edits)

1. **Tone down the manufacturing claim.**
Please soften language on “feasibility of post-fill corrugation / streamlining production,” or move it to Outlook with caveats (repeatability, burst margin, QC, end constraints). See Abstract last lines and Discussion last paragraph.
2. **Make the contribution vs prior “diamond/Yoshimura” work extra explicit.**
In the Introduction/Discussion, add 2–3 sentences: **this work = rings + material nonlinearities**; prior = **diamonds + indentation-driven nonlinearity**. Cite and contrast clearly.
3. **Fix the wavelength prefactor mismatch and report fit quality.**
Fig. 2B text/legend list **A = 6.74 ± 1.78** and **A = 6.47 ± 1.78**. Please make these consistent and report CI / R^2 in the caption or main text.
4. **State what happens on unloading (hysteresis or residual corrugation).**
You say results are **independent of strain/compression rate**. Add one line on unloading: was hysteresis tested? any permanent corrugation? If not tested, please say so. Align wording across Results and Methods.
5. **Model closures: show a small sensitivity.**
 - a) **Axial inextensibility.** After Eq. (9)/(25), acknowledge this strong assumption and, if possible, include a tiny test with small axial stretch to see its effect on w_{max} and ϵ_{max} (SI).
6. **Boundary conditions and end effects.**
You apply **simply supported** ends in the model, but real cans have domed ends and friction; first-buckle location varies between tests. Add one sentence on this limitation and its likely bias on nucleation.
7. **Robustness to constitutive parameters (compact).**
You vary γ_2 (nice). Consider adding a small (γ_1, γ_2) “phase sketch” (even schematic) showing where sequential localized branches exist. This can go to SI near the current sensitivity plots.

Minor presentation

- Check notation/units are consistent (e.g., define Q_L , keep symbols uniform).
- In Fig. 3, consider tiny cartoons next to B–E and label w_{max} , ϵ_{max} once on the panel for readers.

Recommendation: Minor revision.

The core is strong. With the edits above, the paper will be clearer and more balanced.